# Effects of Probiotics as Antibiotics Substitutes on Growth Performance, Serum Biochemical Parameters, Intestinal Morphology, and Barrier Function of Broilers

**DOI:** 10.3390/ani9110985

**Published:** 2019-11-18

**Authors:** Tengfei He, Shenfei Long, Shad Mahfuz, Di Wu, Xi Wang, Xiaoman Wei, Xiangshu Piao

**Affiliations:** State Key Laboratory of Animal Nutrition, College of Animal Science and Technology, China Agricultural University, Beijing 100193, China; hetengfei@cau.edu.cn (T.H.); longshenfei@cau.edu.cn (S.L.); shadmahfuz@yahoo.com (S.M.); superwudee@163.com (D.W.); wang_xi_1998@163.com (X.W.); 13031901582@163.com (X.W.)

**Keywords:** antibiotics, broiler, growth performance, intestinal health, probiotics

## Abstract

**Simple Summary:**

The abuse of antibiotics in animals feed may cause antibiotic-resistant microbes and antibiotic residue in animal products. Probiotics (PB) have been used in the feed industry for several decades due to their beneficial effects on immunity and the growth of livestock and poultry. However, the efficiency of PB on animals varies due to the types and dose of PB. Therefore, investigating the effects of PB (*Bacillus subtilis*, *Bacillus licheniformis*, and *Saccharomyces cerevisiae*) as an antibiotic substitute on growth performance and intestinal health status in broilers is valuable and meaningful.

**Abstract:**

The aim of this study was to investigate the effects of the combination of probiotics replacing antibiotics on growth performance, serum biochemical parameters, intestinal morphology, and expression of tight junction proteins in intestinal mucosa of broilers. A total of 168 Arbor Acres broilers (45.04 ± 0.92 g) were randomly divided into three treatments, with seven replicates per treatment, and eight broilers per replicate. The experiment included phases 1 (d 0 to 21) and 2 (d 21 to 42). The dietary treatments contained a corn soybean meal-based diet (control group; CON); an antibiotic group (basal diet + 75 mg/kg chlortetracycline; CTC), and a probiotics group (basal diet + probiotics (500 mg/kg in phase 1 and 300 mg/kg in phase 2; *Bacillus subtilis* 5 × 10^9^ CFU/g, *Bacillus licheniformis* 2.5 × 10^10^ CFU/g and *Saccharomyces cerevisiae* 1 × 10^9^ CFU/g; PB). The results showed broilers fed PB had improved (*p* < 0.05) feed conversion ratio (FCR) in phase 1 and increased (*p* < 0.05) average daily gain (ADG) in phase 2, as well as improved (*p* < 0.05) ADG and FCR overall (d 0 to 42). The apparent total tract digestibility (ATTD) of dry matter, organic matter, gross energy, and crude protein was increased (*p* < 0.05) in broilers fed PB, while the ATTD of dry matter and organic matter was enhanced in broilers fed CTC compared with CON. Broilers fed PB showed increased (*p* < 0.05) serum total antioxidant capacity concentrations and tended to have higher (*p* = 0.06) level of serum immunoglobulin M in phase 1 compared with CON. These broilers also had increased (*p* < 0.05) level of serum immunoglobulin A in phase 2 in comparison with CON and CTC. Moreover, broilers fed CTC and PB showed increased (*p* = 0.05) villus height to crypt depth ratio in duodenum, as well as higher (*p* < 0.05) mRNA expression of zonula occludens-1 in jejunum compared with CON. In conclusion, dietary supplementation with PB as chlortetracycline substitute could improve the growth performance, nutrient digestibility, serum antioxidant capacity, jejunal mucosal barrier function, and intestinal morphology of broilers.

## 1. Introduction

The wide application of antibiotics has greatly improved the growth performance of livestock and poultry, whereas the abuse of antibiotics in animal feeds may cause antibiotic residue in animal products and the direct selection of antibiotic-resistant microbes, which may cause harm in humans [1]. Broilers, which are one of the fastest growing applications of animal husbandry, face significant problems that impact their growth performance and intestinal health [2]. Therefore, seeking alternatives for in-feed antibiotics for broilers has gained enormous interest currently.

Studies show that herb extracts [3], essential oils [4], and probiotics (PB) [5] could be used as antibiotic substitutes in animals. Among these, PB have been used in feed processing for decades due to their beneficial effects on immune function and growth rate, as well as their low production cost [2]. *Bacillus licheniformis*, which is generally recognized as safe, has been extensively used for a long time in the poultry industry and has demonstrated a positive effect in aiding nutrient digestion and absorption in the host’s body [6,7]. In addition, research has proved that *Bacillus subtilis* improves broiler growth and performance equally as well as antibiotics such as bacitracin methylene disalicylate and avilamycin, and supplementation of *Bacillus subtilis* not only improves broiler performance but also positively impacts villi histomorphometry [8]. These bacteria can also produce digestive enzymes, such as protease, amylase, and lipase, and promote the digestion and absorption of nutrients. Bacterial components, such as cell wall sugar and peptidoglycan, can also promote the growth and development of immune organs in poultry [2]. *Saccharomyces cerevisiae* is a type of anaerobic bacteria, which is rich in protein, nucleic acid, vitamins, polysaccharides, and other nutrients, and its cell wall has a special spatial structure, which can reduce the toxicity of mycotoxins in animals [2]. However, less is known about the effect of the mixture of these three probiotics (*Bacillus subtilis*, *Bacillus licheniformis*, and *Saccharomyces cerevisiae*) on ameliorating impairment of growth performance and intestinal health in broilers.

Therefore, the aim of this study was to explore the effect of dietary inclusion of *Bacillus subtilis, Bacillus licheniformis*, and *Saccharomyces cerevisiae* in broiler diets, on growth performance, nutrient digestibility, serum immunoglobulin, antioxidant function, intestinal barrier function, and intestinal morphology.

## 2. Materials and Methods

The experimental protocols used in this experiment were approved by the Institutional Animal Care and Use Committee of China Agricultural University (Beijing, China) (No. AW09089102-1). The experiment was carried out at the National Feed Engineering Technology Research Center of the Ministry of Agriculture Feed Industry Center Animal Farm (Hebei, China).

### 2.1. Experimental Products

The main components of the PB were *Bacillus subtilis* 5 × 10^9^ CFU/g, *Bacillus licheniformis* 2.5 × 10^10^ CFU/g, and *Saccharomyces cerevisiae* 1 × 10^9^ CFU/g, which were provided by Beijing Smistyle Sci. and Tech. Development Co., Ltd.

### 2.2. Experimental Animals and Design

A total of 168 one-day-old as-hatched Arbor Acres chicks (weighing 45.04 ± 0.92 g) were purchased from Arbor Acres Poultry Breeding Company (Beijing, China). All the broilers were randomly divided into 3 treatments, 7 replicates per treatment, and 8 chickens per replicate. The trial was divided into two phases: phase 1 (day 0 to 21) and 2 (day 21 to 42). The test period was 42 days. The dietary treatments contained a corn soybean meal-based diet (control group, CON); an antibiotic group (basal diet + 75 mg/kg chlortetracycline, CTC), and a probiotics group (basal diet + probiotics (500 mg/kg in phase 1 and 300 mg/kg in phase 2; PB). The feed formulation was based on National Research Council (NRC, 1994) [9] and the formulation is shown in Table 1.

### 2.3. Detection Index and Measuring Method

#### 2.3.1. Growth Performance

The body weight and feed intake of the broilers were registered on day 0, 21, and 42, and the average daily gain (ADG), average daily feed intake (ADFI), and feed conversion ratio (FCR) were calculated.

#### 2.3.2. Nutrient Retention

At day 39–42, 200 g of the excreta sample was collected for each replicate, feathers and the like in the excreta were removed, and the sample was then oven dried at 65 °C for 72 h. All samples were ground to pass through a 1 mm screen (40 mesh) before analysis. Feed or fecal samples were analyzed for dry matter (DM), crude protein (CP), crude fat (EE), and ash according to Association of Official Agricultural Chemists (AOAC, 2012) [10]. The gross energy (GE) in feed and fecal samples were determined by an automatic isoperibol oxygen bomb calorimeter (Parr 1281, Automatic Energy Analyzer; Moline, IL, USA). Organic matter (OM) was calculated as 1 − ash content (DM-base). Nutrient retention was determined by the equation as follows: Apparent total tract digestibility_nutrient_ (ATTD) = 1 − (Cr_diet_ × Nutrient_feces_)/(Cr_feces_ × Nutrient_diet_).

#### 2.3.3. Serum Antioxidant and Immune Function

At day 21 and 42, one broiler chicken with a body weight close to the average was selected for each replicate. A quantity of 4 mL of blood was collected from the wing vein and centrifuged at 3000 r/min for 10 min, and the supernatant was dispensed into a 0.5 mL Eppendorf tube and stored at −80 °C. The contents of serum total antioxidant capacity (T-AOC), superoxide dismutase (SOD), and glutathione peroxidase (GSH-Px) were determined by spectrophotometric methods using a spectrophotometer (Leng Guang SFZ1606017568, Shanghai, China) following the instructions provided by manufacturer (Nanjing Jiancheng Bioengineering Institute, Nanjing, China). The contents of serum malondialdehyde (MDA) were determined using kits from Nanjing Jiancheng Bioengineering Institute (Nanjing, China). The contents of serum immunoglobulin G (IgG), immunoglobulin A (IgA), and immunoglobulin M (IgM) were measured by an ELISA kit (IgG, IgM, and IgA quantitation kit; Bethyl Laboratories, Inc., Montgomery, TX, USA).

#### 2.3.4. Intestinal Morphology

On day 42 of this experiment, two broilers were slaughtered from each replicate. The abdominal cavity was dissected and the intestine was separated. Segments of the mid-duodenum, mid-jejunum, and mid-ileum were taken and rinsed with cold physiological saline (0.9% saline), then immediately stored in 10% buffered formalin. Conventional paraffin embedding, sectioning, HE staining, and six straight and complete fluffs were selected for each section, and the height of the villi and the depth of the crypt corresponding to the villi were determined. The height of random orientated villi and their adjoined crypts were determined with a light microscope using a calibrated eyepiece graticule [11].

#### 2.3.5. The Level of Claudin-1, Occludin and ZO-1 Gene Expression in Jejunal Mucosa

On day 42 of this experiment, the jejunal mucosa was taken from the broilers and then stored in liquid nitrogen. Total RNA extraction was done using Trizol Reagent (TaKaRa, Dalian, China), and the purity and concentration of total RNA were measured by ultraviolet spectrophotometer. Total RNA (1μg) was reverse-transcribed into cDNA using Prime Script RT Reagent Kit (TaKaRa, Dalian, China) according to the direction of the manufacturer’s protocol. The primers were synthesized by TaKaRa Biotechnology (TaKaRa, Dalian, China), which were obtained from the published works of Shao et al and Li et al [12,13], and are shown in Table 2. Real-time PCR was conducted according to Li et al [14].

### 2.4. Statistical Methods

Data was subjected to Analysis of variance (ANOVA) using the GLM procedure of SAS (SAS Institute, 2008) [15]. The replicate was the experimental unit. Significantly different means were separated by Duncan’s multiple range test. Results were expressed as least squares means and SEM. Significance was designated at *p* ≤ 0.05, while a tendency for significance was designated at 0.05 < *p* ≤ 0.10. 

## 3. Results

### 3.1. Growth Performance 

As can be seen from Table 3, dietary supplementation with CTC and PB had no significant effect on the ADFI of broilers compared with CON. In phase 1, broilers fed PB showed improved FCR compared with CON and CTC (*p* < 0.05). In phase 2, broilers fed PB showed improved ADG in comparison with CON (*p* < 0.05) and had no significant difference with CTC. Overall (day 0 to 42), broilers fed PB had improved ADG and FCR compared with CON (*p* < 0.05) and enhanced ADG compared with CTC (*p* < 0.05).

### 3.2. The ATTD of Nutrients

The effects of PB on the ATTD of nutrients in broilers are shown in Table 4. Compared with CON, the DM and OM were increased (*p* < 0.05) in broilers fed PB and CTC. In addition, broilers fed PB also showed enhanced (*p* < 0.05) GE and CP compared with CON.

### 3.3. Serum Antioxidant Status

The effects of PB on the antioxidant status of broilers are shown in Table 5. Compared with CON, broilers fed CTC and PB showed increased serum T-AOC concentration in phase 1 (*p* < 0.05). There was a tendency of enhancing concentration of SOD (*p* = 0.06), GSH-Px (*p* = 0.06), and reducing (*p* = 0.07) level of MDA in serum of broilers fed PB compared with CON in phase 1. In phase 2, broilers fed PB had higher (*p* < 0.05) concentration of GSH-Px and lower (*p* < 0.05) level of MDA in serum in comparison with CON.

### 3.4. Serum Immunoglobulins

The effects of PB on serum immunoglobulins of broilers are shown in Table 6. In phase 1, broilers fed PB (*p* = 0.07) and CON (*p* = 0.06) tended to show enhanced level of IgM compared with CTC, while broilers fed PB increased (*p* < 0.05) level of IgA in phase 2 in comparison with CTC and CON.

### 3.5. Intestinal Morphology

The effects of PB on the intestinal morphology of broilers are shown in Table 7. Compared with CON, the duodenal villus height to crypt depth ratio was significantly increased (*p* < 0.05) in broilers fed CTC and PB. In addition, these broilers tended to showed lower crypt depth in duodenum, as well as higher villus height to crypt depth ratio in ileum compared with CON.

### 3.6. Jejunal Mucosal Barrier Functions

The effects of PB on jejunal mucosal barrier function of broilers are shown in Table 8. Compared with CON, broilers fed CTC and PB showed higher gene expression of zonula occludens-1 (ZO-1) in jejunum (*p* < 0.01), and had no significant effect on the gene expression of claudin-1 and occludin in jejunum.

## 4. Discussion

The current study showed that broilers fed PB improved ADG in phase 2, and ADG and FCR overall. Our results are consistent with the study of Kalia et al. [16], who reported that a diet supplemented with mixed PB could improve the body weight gain and feed efficiency, and decrease mortality in broilers. However, studies conducted by Ahmad et al. [17] and Fathi et al. [18] showed PB had no significant effects on improving FCR. This difference might be due to the variation of survivability of PB in the intestine of broilers and the dose rate of PB used for broilers. The possible reason for the current positive effect on performance could be explained by the *Bacillus subtilis* in PB improving the immune response [19] and the positive effect of PB on modulating the microbiota structure (such as reducing the content of *Salmonella Enteritidis*) [20]. The improvement of performance might also be due to PB increasing nutrient retention (GE, CP, DM, and OM). Research has shown that PB is able to improve the activity of digestive enzymes of animals [21]. Moreover, dietary PB supplementation could produce some metabolites, including organic acids, to enhance the nutrient retention in broilers [22]. The current study showed dietary inclusion of PB, namely, *Bacillus subtilis*, *Bacillus licheniformis*, and *Saccharomyces cerevisiae*, has the same effects as CTC in improving growth performance, which indicates that PB could be a potential antibiotics substitute.

Current research indicates that addition of PB had a positive role on antioxidant functions in broilers. In agreement with our results, Capcarova et al. and Wen et al. [23,24] found that some probiotics could be beneficial in oxidation resistance, scavenging reactive oxygen species, and promoting antioxidant capability. With regard to antioxidant capacity, the endogenous antioxidant defense system in animals also relies on other external sources, such as probiotics, which are the natural source for prevention of the oxidative stress induced by reactive oxygen species [25]. Collectively, this study suggested that PB can possess antioxidant capacity in broilers.

The current study also showed dietary PB supplementation had a positive effect on serum immunoglobulin, which is in agreement with Fathi et al. [18], who reported improving effects of PB on IgM and cell-mediated immunity. The reason may be that *Bacillus subtilis* had a positive effect on enhancing antibodies against the Newcastle disease of broiler chicks [19]. PB *Bacillus subtilis* could also enhance humoral immune responses and stimulate the host’s mucosal immune system by interacting with intestinal epithelial cells in broilers [26]. The mechanism of PB on the immunity of broilers may also result because PB can protect animals from pathogen colonization by competing for epithelial binding sites and nutrients, strengthening the intestinal immune response, and producing antimicrobial bacteriocins. [22]

The current study showed that dietary PB supplementation can increase the ratio of villus height to crypt depth, which indicates that PB can promote the development of the absorptive surface of duodenum and ileum in broilers. This might be due to the beneficial bacteria in PB, which may improve crypt cell proliferation in the small intestine, and thus help increase the growth rate in broilers [17]. In addition, the *Bacillus licheniformis* in PB can colonize and form niches in the small intestine, which positively protects the villi from pathogens and improves the growth of villi [27]. However, Sohail et al. [28] found that PB had no effect on stress-induced injury in the intestinal morphology of 42-day-old chickens, which might be due to the variation of types and amounts of PB used in different studies. Moreover, the improvement of intestinal morphology and integrated intestinal barrier are important for epithelial cell function, which might be the reason for the improved ATTD of nutrients [29].

The function of the intestinal barrier and the absorption of nutrients can be directly affected by the damage of the mucosal epithelium, and PB can regulate intestinal immunity and tight junction protein mRNA expression of broilers [30]. Current research indicates that the addition of PB to diets can promote the gene expression of ZO-1 in jejunal mucosa of broilers and improve the jejunal mucosal barrier function of broilers. PB in diets can decrease the feed weight gain ratio and intestinal coliform, and can also increase the duodenal villus height to crypt depth ratio. These results suggest that the supplementation of a PB mixture in the diet can effectively improve part of the intestinal barrier function. PB has been shown to be adherent to the intestinal epithelium, resistant to acidic conditions, and capable of antagonizing and competitively eliminating certain pathogens in vivo [31]. In contrast, the PB mixture used in this study consisted of *Bacillus licheniformis*, *Bacillus subtilis*, and *Saccharomyces cerevisiae*. *Bacillus licheniformis* and *Bacillus subtilis* are aerobic bacteria that use oxygen in the intestine to provide an anaerobic environment for the colonization of anaerobic bacteria, such as *Lactobacilli* and *Bifidobacteria*. Therefore, these lactic acid-producing bacteria produce a more acidic environment, which impairs the growth of opportunistic pathogens [32].

## 5. Conclusions

The results of this experiment showed that the addition of probiotics (500 mg/kg in phase 1, 300 mg/kg in phase 2) could improve broilers’ growth performance, nutrient retention, and serum antioxidant capacity, and improve their intestinal health via improving jejunal mucosal barrier function and intestinal morphology. The results indicated that the current probiotics could be used as a chlortetracycline substitute in the diet of broiler chickens.

## Figures and Tables

**Table 1 animals-09-00985-t001:** Composition and nutrient levels of basal diets (%, as-fed basis).

Ingredients	Day 0 to 21	Day 21 to 42
Corn	58.17	64.26
Soybean meal, 43%	30.44	24.05
Corn gluten meal	2.00	2.50
Fish meal, 64.6%	2.00	2.00
Soybean oil	3.38	3.60
Dicalcium phosphate	1.50	1.04
Limestone	1.30	1.35
Salt	0.30	0.30
L-lysine HCl, 78%	0.01	0.08
DL-Methionine, 98%	0.14	0.04
L-Threonine, 98%	0.01	0.03
Chromic oxide	0.25	0.25
Vitamin-mineral premix ^1^	0.50	0.50
Total	100.00	100.00
Calculated nutrient levels ^2^		
Metabolizable energy, MJ/kg	12.76	13.17
Crude protein	21.00	19.00
Calcium	1.00	0.90
Available phosphorus	0.45	0.35
Standardized ileal digestible lysine	0.86	0.73
Standardized ileal digestible methionine	0.30	0.28
Standardized ileal digestible threonine	0.63	0.56
Standardized ileal digestible tryptophan	0.28	0.24

^1.^ Vitamin A, 11,000 IU; vitamin D, 3025 IU; vitamin E, 22 mg; vitamin K_3_, 2.2 mg; thiamine, 1.65 mg; riboflavin, 6.6 mg; pyridoxine, 3.3 mg; cobalamin, 17.6 μg; nicotinic acid, 22 mg; pantothenic acid, 13.2 mg; folic acid, 0.33 mg; biotin, 88 μg; choline chloride, 500 mg; iron, 48 mg; zinc, 96.6 mg; manganese, 101.76 mg; copper, 10 mg; selenium, 0.05 mg; iodine, 0.96 mg; cobalt, 0.3 mg. ^2.^ Crude protein was the analyzed value. Other values were calculated.

**Table 2 animals-09-00985-t002:** Sequences of the primers used for the determination of gene expression levels.

Genes	Primer Sequence (5′-3′)	Gene Accession No.	References
Claudin-1	F: TGGAGGATGACCAGGTGAAGA	NM_001013611.2	[11]
	R: CGAGCCACTCTGTTGCCATA		
Occludin	F: TCATCGCCTCCATCGTCTAC	NM_205128.1	[11]
	R: TCTTACTGCGCGTCTTCTGG		
ZO-1	F: TGTAGCCACAGCAAGAGGTG	XM 413773.4	[12]
	R: CTGGAATGGCTCCTTGTGGT		

ZO-1: zona occludens-1.

**Table 3 animals-09-00985-t003:** Effects of probiotics on growth performance of broilers ^1^.

Item	CON	CTC	PB	SEM	*p*-Value			
					Treatment ^2^	CON vs. CTC	CON vs. PB	CTC vs. PB
day 0 to 21								
Average daily gain, g	27.32	26.96	28.41	0.47	0.15	0.45	0.56	0.11
Average daily feed intake, g	36.99	36.42	35.93	0.62	0.76	0.73	0.49	0.80
Feed conversion ratio	1.35 ^a^	1.35 ^a^	1.26 ^b^	0.02	0.02	0.96	0.01	0.02
day 21 to 42								
Average daily gain, g	68.85 ^b^	71.44 ^a,b^	76.62 ^a^	1.12	<0.01	0.12	<0.01	0.34
Average daily feed intake, g	105.87	105.05	112.35	2.55	0.12	0.89	0.16	0.11
Feed conversion ratio	1.54	1.47	1.46	0.03	0.24	0.43	0.36	0.85
day 0 to 42								
Average daily gain, g	48.08 ^b^	49.20 ^b^	52.52 ^a^	0.52	<0.01	0.79	<0.01	<0.01
Average daily feed intake, g	71.43	70.73	74.14	1.10	0.09	0.82	0.10	0.08
Feed conversion ratio	1.49 ^a^	1.44 ^a,b^	1.41 ^b^	0.02	0.04	0.23	0.02	0.15

^1^ CON, control; CTC, chlorotetracycline (75 mg/kg); PB, probiotic (500 mg/kg in phase 1, 300 mg/kg in phase 2); SEM, standard error of the mean. ^2^ Treatment, the specific *p* value of the diet effect in the ANOVAs analysis. ^a,b^ values in the same row with different letters are significantly different at *p* < 0.05.

**Table 4 animals-09-00985-t004:** Effects of probiotics on the nutrient retention of broilers (%, day 42) ^1^.

Items	CON	CTC	PB	SEM	*p*-Value			
					Treatment ^2^	CON vs. CTC	CON vs. PB	CTC vs. PB
Dry matter	73.80 ^b^	75.42 ^a^	75.92 ^a^	0.30	<0.01	<0.01	<0.01	0.72
Gross energy	76.58 ^b^	78.07 ^a,b^	78.38 ^a^	0.39	0.03	0.16	0.01	0.84
Crude protein	64.04 ^b^	66.83 ^a,b^	67.95 ^a^	0.95	0.06	0.09	0.03	0.42
Ether extract	91.57	93.05	91.71	0.68	0.31	0.52	0.66	0.13
Organic matter	76.49 ^b^	77.93 ^a^	78.33 ^a^	0.30	0.01	0.02	<0.01	0.63

^1^ CON, control; CTC, chlorotetracycline (75 mg/kg); PB, probiotic (500 mg/kg in phase 1, 300 mg/kg in phase 2); SEM, standard error of the mean. ^2^ Treatment, the specific *p* value of the diet effect in the ANOVAs analysis. ^a,b^ values in the same row with different letters are significantly different at *p* < 0.05.

**Table 5 animals-09-00985-t005:** Effects of probiotics on the serum antioxidant function of broilers ^1^.

Items	CON	CTC	PB	SEM	*p*-Value			
					Treatment ^2^	CON vs. CTC	CON vs. PB	CTC vs. PB
d 0 to 21								
T-AOC (U/ml)	1.00 ^b^	2.50 ^a^	2.37 ^a^	0.40	0.05	0.02	0.04	0.53
SOD (U/ml)	55.21	56.61	57.13	0.55	0.08	0.24	0.06	0.36
GSH-Px (μmol/L)	27.39	30.16	36.02	2.56	0.09	0.28	0.06	0.14
MDA (nmol/ml)	7.64	7.10	6.90	0.22	0.10	0.25	0.07	0.44
d 21 to 42								
T-AOC (U/ml)	3.33	2.99	3.35	0.52	0.77	0.41	0.89	0.37
SOD (U/ml)	87.24	75.05	87.99	4.35	0.12	0.11	0.74	0.14
GSH-Px (μmol/L)	19.42 ^b^	23.92 ^a,b^	26.75 ^a^	1.90	0.06	0.23	0.04	0.46
MDA (nmol/ml)	7.58 ^a^	6.46 ^a,b^	6.09 ^b^	0.41	0.07	0.06	0.03	0.75

^1^ CON, control; CTC, chlorotetracycline (75 mg/kg); PB, probiotic (500 mg/kg in phase 1, 300 mg/kg in phase 2); SEM, standard error of the mean. ^2^ Treatment, the specific *p* value of the diet effect in the ANOVAs analysis. ^a,b^ values in the same row with different letters are significantly different at *p* < 0.05.

**Table 6 animals-09-00985-t006:** Effects of probiotics on the serum immunoglobulins function of broilers (ug/mL) ^1^.

Items	CON	CTC	PB	SEM	*p*-Value			
					Treatment ^2^	CON vs. CTC	CON vs. PB	CTC vs. PB
day 0 to 21								
Immunoglobulin A	4.24	4.10	4.40	0.12	0.24	0.68	0.57	0.32
Immunoglobulin G	2.27	2.23	2.28	0.03	0.45	0.43	0.51	0.39
Immunoglobulin M	1.71	1.65	1.70	0.02	0.07	0.06	0.88	0.07
day 21 to 42								
Immunoglobulin A	4.27 ^b^	4.25 ^b^	4.98 ^a^	0.22	0.07	0.76	0.04	0.04
Immunoglobulin G	2.28	2.25	2.51	0.14	0.39	0.65	0.22	0.18
Immunoglobulin M	1.68	1.68	1.91	0.14	0.44	0.83	0.26	0.24

^1^ CON, control; CTC, chlorotetracycline (75 mg/kg); PB, probiotic (500 mg/kg in phase 1, 300 mg/kg in phase 2); SEM, standard error of the mean. ^2^ Treatment, the specific *p* value of the diet effect in the ANOVAs analysis. ^a,b^ values in the same row with different letters are significantly different at *p* < 0.05.

**Table 7 animals-09-00985-t007:** Effects of probiotics on intestinal morphology of broilers (day 42) ^1^.

Items	CON	CTC	PB	SEM	*p*-Value			
					Treatment ^2^	CON vs. CTC	CON vs. PB	CTC vs. PB
Duodenum								
Villus height (μm)	1905	2234	2134	128	0.26	0.14	0.21	0.37
Crypt depth (μm)	269	172	181	26	0.07	0.06	0.09	0.26
Villus height/ Crypt depth	7.95 ^b^	13.10 ^a^	12.04 ^a^	1.19	0.05	0.03	0.04	0.64
Jejunum								
Villus height (μm)	1372	1561	1360	95	0.32	0.14	0.72	0.11
Crypt depth (μm)	253	195	177	35	0.34	0.16	0.11	0.61
Villus height/ Crypt depth	6.00	8.19	8.25	1.10	0.33	0.45	0.28	0.76
Ileum								
Villus height (μm)	1204	1103	1097	84	0.62	0.32	0.25	0.84
Crypt depth (μm)	159	138	105	20	0.25	0.36	0.14	0.62
Villus height/ Crypt depth	8.14	8.27	10.58	0.72	0.09	0.62	0.06	0.13

^1^ CON, control; CTC, chlorotetracycline (75 mg/kg); PB, probiotic (500 mg/kg in phase 1, 300 mg/kg in phase 2); SEM, standard error of the mean. ^2^ Treatment, the specific *p* value of the diet effect in the ANOVAs analysis. ^a,b^ values in the same row with different letters are significantly different at *p* < 0.05.

**Table 8 animals-09-00985-t008:** Effects of probiotics on gene expression levels of claudin-1, occludin and ZO-1 genes in the jejunum barrier function of broilers (day 42) ^1^.

Items	CON	CTC	PB	SEM	*p*-Value			
					Treatment ^2^	CON vs. CTC	CON vs. PB	CTC vs. PB
Claudin-1	1.00	0.93	0.53	0.18	0.18	0.72	0.12	0.14
Occludin	0.99	1.41	0.98	0.31	0.27	0.13	0.87	0.11
ZO-1	1.02 a,b	5.98 a	5.66 a	0.57	<0.01	<0.01	<0.01	0.92

^1^ CON, control; CTC, chlorotetracycline (75 mg/kg); PB, probiotic (500 mg/kg in phase 1, 300 mg/kg in phase 2); SEM, standard error of the mean. ^2^ Treatment, the specific *p* value of the diet effect in the ANOVAs analysis. ^a,b^ values in the same row with different letters are significantly different at *p* < 0.05.

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
