# Peer review of "Effects of Probiotics as Antibiotics Substitutes on Growth Performance, Serum Biochemical Parameters, Intestinal Morphology, and Barrier Function of Broilers"

_animals, 2019, doi:10.3390/ani9110985_

Round 1

Reviewer 1 Report

General: the work present is valuable and worthy of publication. However, there are many deficiencies in the manuscript that the authors need to address before the manuscript can be considered further. These are not fatal flaws in themselves, but are significant enough that the reviewer thought that the authors should adequately address those. This will include interpretation of results and its presentation, a few statistical issues that can be fixed with appropriate statistical tools, and a large number of language usage issues. 

Title: Perhaps it’s an overreach to refer to probiotics as antibiotic replacer – perhaps better to refer to them as alternatives, substitutes etc

L25: perhaps this should be written as day 0 to 21, day 21 to 42, etc

L64: which organism is this referring to? The authors stated earlier on that much work has been doine on Bacillus…

L82: if they were divided to treatments “according to body weight”, then they were not randomly divided. This is like a randomized complete block design, in which birds were allocated to treatments and blocks according to weight, using body weight as blocking criterion

L83: trial…

L83: please see previous comments on age,0 to 21 etc

L86: why did the authors use NRC guide instead of the breed recommendation?

Table 1:

Do these values have to be centered?

Please add up the items to 100

Is this digestible, or available, or non-phytate P? NRC doesn’t have requirement values for digestible P

L96: please rephrase this sentence. Feed weight is not for the broilers

L100: excreta, not stool

L100: replicates, not repetition

L102: what is resurgence? Maybe need to use a different expression

L102: does pulverize mean grinding?

L102: is digestion different from utilization rate? How were they calculated – the utilization rate?

L107: replicates, not repeated? Perhaps the authors could use the help of someone to improve the flow of their manuscript

L109: store at -80C is not a complete sentence

L109: needs to be rewritten to be a report rather than instruction

L113: the jugular vein was bled to death – this doesn’t make any sense

L114-6: please rewrite this sentence

L118-9: what equipment was used for this?

L121-2: please see comment on line 113

L122: the sentence in line 122 is not meaningful

L123: the extraction was done using Trizol…

L134: significantly different means were separated…

L148: digestibility is measured at the ileal level; therefore if these were measured at the total tract, they should be referred to as “retention”

Table 4: should the AME be calculated, rather than GE utilization?

L152: the p-value of 0.06 is for overall p-value and does not represent the comparison of PB vs. CON alone. Perhaps there was a trend for improvement in CP retention in diet CTC diet as well.

L160-1: see previous comment on line 152

L162-4: see previous comment for line 152

L190-3: I don’t know if I missed it, was any housekeeping gene used here? please emphasize that in the materials and methods section

L201: …which is consistent with…

L207: improving intestinal bacteria structure? What does that mean?

L200-25: I think the authors spent too much time talking about findings of others, which are rather routine, instead of explaining the significance of their own  findings

L227: no digestive enzyme activities were measured here, so this conclusion is unfounded

L228-32: how is this related to nutrient digestibility?

L235-8: again, how’s this related to the model used in the current experiment?

L239: this has to do with the development of the absorptive surface of the intestine, rather than the development of the intestine itself. Please rephrase

L252-7: perhaps the authors can link these with their observation for the effect of antibiotics on the same genes etc.

L267: I think the author should rephrase this section to indicate the possible mechanisms, from their own study, by which PB could exert its beneficial effects.

L299-301: in this reference, what are the page number of the referenced article?

L302-64: these seem to be formatted different from the previous lines, please check

L312: what are the inclusive pages

L332: same as line 312

L343: same as line 332

Author Response

Revision Note, (List of modification) Date: 2019-11-10 (y-m-d)

Manuscript ID: animals-634279-RI.

Dear Sir

Good day. Thank you very much for your kind consideration with our submitted article and offering us the further opportunity to submit the revised manuscript. Please find here the point to point expert reviewer’s and editor’s comments with necessary changes as per suggested with this attached file, and the amendments are highlighted in red in the revised manuscript. We have revised our manuscript for language and grammar checked by a native English speaker working in our University. We do thanks to skilled reviewers, academic editors and editorial board members as well for their critical evaluation to make the manuscript more effective for review process in Animals Journal.

Many thanks.

Sincerely yours,

Prof. Dr. Xiang Shu Piao,

State Key laboratory of Animal Nutrition, College of Animal Science and Technology, China Agricultural University, Beijing 100193, China

Corresponding Author,

Email: [email protected]; Tel.:+86-1062733588/Fax.: +86-1062733688

Comments from Reviewer 1

Comment 1: Title: Perhaps it’s an overreach to refer to probiotics as antibiotic replacer – perhaps better to refer to them as alternatives, substitutes etc

Response 1: Thanks, we have changed the title into “Effects of Probiotics as Antibiotics Substitutes on Growth Performance, Serum Biochemical Parameters, Intestinal Morphology and Barrier Function of Broilers”

Comment 2: L25: perhaps this should be written as day 0 to 21, day 21 to 42, etc

Response 2: Thanks, we have corrected this kind of mistake in this manuscript.

Comment 3: L64: which organism is this referring to? The authors stated earlier on that much work has been done on Bacillus…

Response 3: Thanks, the organism refers the mixture of these three probiotics (Bacillus subtilis, Bacillus licheniformis, and Saccharomyces cerevisiae), and we also have rewritten this part.

Comment 4: L82: If they were divided to treatments “according to body weight”, then they were not randomly divided. This is like a randomized complete block design, in which birds were allocated to treatments and blocks according to weight, using body weight as blocking criterion

Response 4: Thanks, they were randomly divided into 3 treatments, and did not according to the body weight, and we have corrected this part.

Comment 5: L83: trial…

Response 5: Thanks, we have corrected this part.

Comment 6: L83: Please see previous comments on age,0 to 21 etc

Response 6: Thanks, we have corrected this part.

Comment 7: L86: why did the authors use NRC guide instead of the breed recommendation?

Response 7: Thanks for your advice, we will consider using the breed recommendation in next experiment. In the current trail, we refer to NRC (1994) since it is recommended standard in poultry science field.

Comment 8: Table 1: Do these values have to be centered?

Response 8: Thanks, this is the requirement of this journal.

Comment 9: Please add up the items to 100

Response 9: Thanks, we have added up the items to 100.

Comment 10: Is this digestible, or available, or non-phytate P? NRC doesn’t have requirement values for digestible P

Response 10: Thanks, it is the available phosphorus and we have corrected it

Comment 11: L96: please rephrase this sentence. Feed weight is not for the broilers

Response 11: Thanks, we have rephrased this sentence.

Comment 12: L100: excreta, not stool

Response 12: Thanks, we have corrected this part.

Comment 13: L100: replicates, not repetition

Response 13: Thanks, we have corrected this part.

Comment 14: L102: what is resurgence? Maybe need to use a different expression

Response 14: Thanks, we have rephrased this sentence

Comment 15: L102: does pulverize mean grinding?

Response 15: Thanks, the pulverize mean grinding, and we have corrected it.

Comment 16: L102: is digestion different from utilization rate? How were they calculated – the utilization rate?

Response 16: Thanks, this part measured at the total tract and should be referred to as “retention”. We also have added the calculation method in Materials and Methods part.

Comment 17: L107: replicates, not repeated? Perhaps the authors could use the help of someone to improve the flow of their manuscript

Response 17: Thanks, we have corrected this part, and we have invited someone with high level of English ability to help us improve the flow of this manuscript.

Comment 18: L109: store at -80C is not a complete sentence

Response 18: Thanks, we have rephrased this sentence

Comment 19: L109: needs to be rewritten to be a report rather than instruction

Response 19: Thanks, we have rewritten this part.

Comment 20: L113: the jugular vein was bled to death – this doesn’t make any sense

Response 20: Thanks, we have rephrased this sentence.

Comment 21: L114-6: please rewrite this sentence

Response 21: Thanks, we have rewritten this sentence

Comment 22: L118-9: what equipment was used for this?

Response 22: Thanks, we have rephrased this part, and the height of random orientated villi and their adjoined crypts were determined with light microscope using a calibrated eyepiece graticule

Comment 23: L121-2: please see comment on line 113

Response 23: Thanks, we have corrected this part.

Comment 24: L122: the sentence in line 122 is not meaningful

Response 24: Thanks, we have corrected this part.

Comment 25: L123: the extraction was done using Trizol…

Response 25: Thanks, we have corrected this part.

Comment 26: L134: significantly different means were separated…

Response 26: Thanks, we have corrected this part.

Comment 27: L148: digestibility is measured at the ileal level; therefore if these were measured at the total tract, they should be referred to as “retention”

Response 27: Thanks, we have corrected this part.

Comment 28: Table 4: should the AME be calculated, rather than GE utilization?

Response 28: Thanks, the ATTD shall be GE, and we have corrected this part

Comment 29: L152: the p-value of 0.06 is for overall p-value and does not represent the comparison of PB vs. CON alone. Perhaps there was a trend for improvement in CP retention in diet CTC diet as well.

Response 29: Thanks, we have compared the difference between PB vs. CON, PB vs. CTC, and CTC vs. CON alone, and have rewritten the tables and the related results.

Comment 30: L160-1: see previous comment on line 152

Response 30: Thanks, we have compared the difference between PB vs. CON, PB vs. CTC, and CTC vs. CON alone, and have rewritten the tables and the related results.

Comment 31: L162-4: see previous comment for line 152

Response 31: Thanks, we have compared the difference between PB vs. CON, PB vs. CTC, and CTC vs. CON alone, and have rewritten the tables and the related results.

Comment 32: L190-3: I don’t know if I missed it, was any housekeeping gene used here? please emphasize that in the materials and methods section

Response 32: Thanks, there was no any housekeeping gene used here.

Comment 33: L201: …which is consistent with…

Response 33: Thanks, we have corrected this part.

Comment 34: L207: improving intestinal bacteria structure? What does that mean?

Response 34: Thanks, we have rephrased this part, and it means that probiotic beneficially affect the host by improving the properties of the indigenous microbiota. (Jadhav at al., 2015).

Comment 35: L200-25: I think the authors spent too much time talking about findings of others, which are rather routine, instead of explaining the significance of their own findings

Response 35: Thanks for your comments, and we have deleted some findings of others and explained more on the significant results of the findings.

Comment 36: L227: no digestive enzyme activities were measured here, so this conclusion is unfounded

Response 36: Thanks for your comments, and we have rephrased this part.

Comment 37: L228-32: how is this related to nutrient digestibility?

Response 37: Thanks for your comments, and we have rephrased this part

Comment 38: L235-8: again, how’s this related to the model used in the current experiment?

Response 38: Thanks for your comments, and we have rephrased this part

Comment 39: L239: this has to do with the development of the absorptive surface of the intestine, rather than the development of the intestine itself. Please rephrase

Response 39: Thanks, we have corrected this part.

Comment 40: L252-7: perhaps the authors can link these with their observation for the effect of antibiotics on the same genes etc.

Response 40: we have corrected this part.

Comment 41: L267: I think the author should rephrase this section to indicate the possible mechanisms, from their own study, by which PB could exert its beneficial effects.

Response 41: we have corrected this part.

Comment 42: L299-301: in this reference, what are the page number of the referenced article?

Response 42: Thanks, we did not find the exactly page number of this referenced article, and we have added its DOI in the end.

Comment 43: L302-64: these seem to be formatted different from the previous lines, please check

Response 43: Thanks, we have corrected this part.

Comment 44: L312: what are the inclusive pages

Response 44: Thanks, we did not find the exactly page number of this referenced article, and we have added its DOI in the end.

Comment 45: L332: same as line 312

Response 45: Thanks, we did not find the exactly page number of this referenced article, and we have added its DOI in the end.

Comment 46: L343: same as line 332

Response 46: Thanks, we did not find the exactly page number of this referenced article, and we have added its DOI in the end.

Reviewer 2 Report

Mayor comments:

The manuscript need an extensive revision of English

Abstract: Please resume results in order to simplify the abstract. Some abbreviation need to be first explained to help readers from other research areas.

Line 72: Please add the number assigned by the ethical committee

Results: Results are not well explained. Please add the exact p values of the main effects and of the specific comparisons. There is more information relevant in tables that what you explain in the text and they need to be addressed.

References. Most of the author are from your country. There is several information of this field, please add author from other countries and universities.

Minor comments:

Keywords. Please put the words in alphabetic order.

Line 51: Studies show herb extracts è Studies show that herb extracts

Line 55: Research showed that supplementing feed water è Research showed that supplementing feed or water

Line 56: Bacillus subtilis can reproduce?

Line 66: Therefore, the aim of this study was dietary inclusionè Therefore, the aim of this study was to explore the effect of dietary inclusion

Line 80: Never start a sentence with a number

Line 83: The trail is divided into two phasesè The trail was divided into two phases

Line 86: The feed formulation is basedè Line 83: The feed formulation was based

Line 86: were weighed in the trialsè were registered on …

Line 121-122: At d 42 of this experiment, two broilers were randomly taken from each replicate, and the jugular 122 vein was bled to death. The same broilers than before? Please explain

Line 134: pen was the experimental unit. I wandering if 7 replicates per treatment is enough to observed the differences that you expected?

Line 139-140: No effect in general (1-42d)? Please explain better. No significant effect of the diet over ADFI?

Line 139-143: Please add the exact p values of the main effect and of the specific comparisons. There is more information relevant in tables that what you explain in the text.

Line 149-152: Please add the exact p values of the main effect and of the specific comparisons. There is more information relevant in tables that what you explain in the text.

Line 158-164: Please add the exact p values of the main effect and of the specific comparisons. There is more information relevant in tables that what you explain in the text.

Line 171-174: Please add the exact p values of the main effect and of the specific comparisons. There is more information relevant in tables that what you explain in the text.

Line 180-184: Please add the exact p values of the main effect and of the specific comparisons. There is more information relevant in tables that what you explain in the text.

Line 190-193: Please add the exact p values of the main effect and of the specific comparisons. There is more information relevant in tables that what you explain in the text.

Author Response

Revision Note, (List of modification) Date: 2019-11-10 (y-m-d)

Manuscript ID: animals-634279-RI.

Dear Sir

Good day. Thank you very much for your kind consideration with our submitted article and offering us the further opportunity to submit the revised manuscript. Please find here the point to point expert reviewer’s and editor’s comments with necessary changes as per suggested with this attached file, and the amendments are highlighted in red in the revised manuscript. We have revised our manuscript for language and grammar checked by a native English speaker working in our University. We do thanks to skilled reviewers, academic editors and editorial board members as well for their critical evaluation to make the manuscript more effective for review process in Animals Journal.

Many thanks.

Sincerely yours,

Prof. Dr. Xiang Shu Piao,

State Key laboratory of Animal Nutrition, College of Animal Science and Technology, China Agricultural University, Beijing 100193, China

Corresponding Author,

Email: [email protected]; Tel.:+86-1062733588/Fax.: +86-1062733688

Comments from Reviewer 2

Mayor comments 1: The manuscript need an extensive revision of English

Response 1: Thanks, we have invited someone with great English ability to help us improve the flow of this manuscript.

Comment 2: Abstract: Please resume results in order to simplify the abstract. Some abbreviation need to be first explained to help readers from other research areas.

Response 2: Thanks, we have corrected this part.

Comment 3: Line 72: Please add the number assigned by the ethical committee

Response 3: We have added this No. AW09089102-1.

Comment 4: Results: Results are not well explained. Please add the exact p values of the main effects and of the specific comparisons. There is more information relevant in tables that what you explain in the text and they need to be addressed.

Response 4: We have corrected this part.

Comment 5: References. Most of the author are from your country. There is several information of this field, please add author from other countries and universities.

Response 5: We have added more references conducted abroad.

Comment 6: Keywords. Please put the words in alphabetic order.

Response 6: Thanks, we have putted the key words in alphabetic order.

Comment 7: Line 51: Studies show herb extracts è Studies show that herb extracts

Response 7: Thanks, we have corrected this part.

Comment 8: Line 55: Research showed that supplementing feed water è Research showed that supplementing feed or water

Response 8: Thanks, we have corrected this part.

Comment 9: Line 56: Bacillus subtilis can reproduce?

Response 9: Thanks for your advice, we have rephrased this part, and Bacillus subtilis can reproduce and consume large amounts of oxygen in the intestinal tract of animals

Comment 10: Line 66: Therefore, the aim of this study was dietary inclusion. Therefore, the aim of this study was to explore the effect of dietary inclusion.

Response 10: Thanks, we have corrected this part.

Comment 11: Line 80: Never start a sentence with a number

Response 11: Thanks, we have corrected this kind of mistake.

Comment 12: Line 83: The trail is divided into two phases è The trail was divided into two phases

Response 12: Thanks, we have corrected this part.

Comment 13: Line 86: The feed formulation is based è

Response 13: Thanks, we have corrected this part.

Comment 14: Line 83: The feed formulation was based

Response 14: Thanks, we have corrected this part.

Comment 15: Line 86: were weighed in the trials è were registered on …

Response 15: Thanks, we have corrected this part.

Comment 16: Line 121-122: At d 42 of this experiment, two broilers were randomly taken from each replicate, and the jugular 122 vein was bled to death. The same broilers than before? Please explain

Response 16: Thanks, they were the same broilers and we have rewritten this part.

Comment 17: Line 134: pen was the experimental unit. I wandering if 7 replicates per treatment is enough to observed the differences that you expected?

Response 17: Thanks, and in the design, 7 replicates are the least for the data analysis

Comment 18: Line 139-140: No effect in general (1-42d)? Please explain better. No significant effect of the diet over ADFI?

Response 18: Thanks, we have explained in a better way.

Comment 19: Line 139-143: Please add the exact p values of the main effect and of the specific comparisons. There is more information relevant in tables that what you explain in the text.

Response 19: Thanks, we have compared the difference between PB vs. CON, PB vs. CTC, and CTC vs. CON alone, and added the exact p values and also rewritten the tables and related results.

Comment 20: Line 149-152: Please add the exact p values of the main effect and of the specific comparisons. There is more information relevant in tables that what you explain in the text.

Response 20: Thanks, we have compared the difference between PB vs. CON, PB vs. CTC, and CTC vs. CON alone, and added the exact p values and also rewritten the tables and related results.

Comment 21: Line 158-164: Please add the exact p values of the main effect and of the specific comparisons. There is more information relevant in tables that what you explain in the text.

Response 21: Thanks, we have compared the difference between PB vs. CON, PB vs. CTC, and CTC vs. CON alone, and added the exact p values and also rewritten the tables and related results.

Comment 22: Line 171-174: Please add the exact p values of the main effect and of the specific comparisons. There is more information relevant in tables that what you explain in the text.

Response 22: Thanks, we have compared the difference between PB vs. CON, PB vs. CTC, and CTC vs. CON alone, and added the exact p values and also rewritten the tables and related results.

Comment 23: Line 180-184: Please add the exact p values of the main effect and of the specific comparisons. There is more information relevant in tables that what you explain in the text.

Response 23: Thanks, we have compared the difference between PB vs. CON, PB vs. CTC, and CTC vs. CON alone, and added the exact p values and also rewritten the tables and related results.

Comment 24: Line 190-193: Please add the exact p values of the main effect and of the specific comparisons. There is more information relevant in tables that what you explain in the text.

Response 24: Thanks, we have compared the difference between PB vs. CON, PB vs. CTC, and CTC vs. CON alone, and added the exact p values and also rewritten the tables and related results.

Round 2

Reviewer 2 Report

Mayor comments:

Abstract: Please resume results in order to simplify the abstract. First describe main effects of treatment and after that specific comparisons

Results: Please add the exact p values of the main effects and of the specific comparisons in the text. In figures different letters could be added to describe means differences.

References. Most of the author are from your country. There is several information of this field, please add author from other countries and universities.

Minor comments:

Line 152-157: No effect in general (1-42d)? Please explain better. No significant effect of the diet over ADFI? Where is the specific p value of the diet effect in the ANOVAs analysis?

Line 162-164: Please add the exact p values of the main effect (diet) and of the specific comparisons. In figures different letters could be added to describe means differences.

Line 169-174: Please add the exact p values of the main effect (diet) and of the specific comparisons. In figures different letters could be added to describe means differences.

Line 184-186: Please add the exact p values of the main effect (diet) and of the specific comparisons. In figures different letters could be added to describe means differences.

Line 191-194: Please add the exact p values of the main effect (diet) and of the specific comparisons. In figures different letters could be added to describe means differences.

Line 201-204: Please add the exact p values of the main effect (diet) and of the specific comparisons. In figures different letters could be added to describe means differences.

Author Response

Revision Note, (List of modification) Date: 2019-11-13 (y-m-d)

Manuscript ID: animals-634279-RI.

Dear Sir

Good day. Thank you very much for your kind consideration with our submitted article and offering us the further opportunity to submit the revised manuscript. Please find here the point to point expert reviewer’s and editor’s comments with necessary changes as per suggested with this attached file, and the amendments are highlighted in red in the revised manuscript. We have revised our manuscript for language and grammar checked by a native English speaker working in our University. We do thanks to skilled reviewers, academic editors and editorial board members as well for their critical evaluation to make the manuscript more effective for review process in Animals Journal.

Many thanks.

Sincerely yours,

Prof. Dr. Xiang Shu Piao,

State Key laboratory of Animal Nutrition, College of Animal Science and Technology, China Agricultural University, Beijing 100193, China

Corresponding Author,

Email: [email protected]; Tel.:+86-1062733588/Fax.: +86-1062733688

Mayor comments:

Abstract: Please resume results in order to simplify the abstract. First describe main effects of treatment and after that specific comparisons

Response: Thanks. We have simplified the abstract according to your suggestion.

Results: Please add the exact p values of the main effects and of the specific comparisons in the text. In figures different letters could be added to describe means differences.

Response: Thanks. We have added the exact p values and different letters to describe means differences in figures.

References. Most of the author are from your country. There is several information of this field, please add author from other countries and universities.

Response: Thanks. We have added some references from other countries and universities.

Minor comments:

Line 152-157: No effect in general (1-42d)? Please explain better. No significant effect of the diet over ADFI? Where is the specific p value of the diet effect in the ANOVAs analysis?

Response: Thanks. We have added the exact p values. And the specific P value for diets effect was 0.09, that was the treatment effect for feed intake overall period. (Table 3, d 0-42,); However, during 1-42 d ADG and FCR were significant. (Table 3). Treatment effect is the specific diet effect (P) in the ANOVA table.

Line 162-164: Please add the exact p values of the main effect (diet) and of the specific comparisons. In figures different letters could be added to describe means differences.

Response: Thanks. We have added the exact p values and different letters to describe means differences in figures.

Line 169-174: Please add the exact p values of the main effect (diet) and of the specific comparisons. In figures different letters could be added to describe means differences.

Response: Thanks. We have added the exact p values and different letters to describe means differences in figures.

Line 184-186: Please add the exact p values of the main effect (diet) and of the specific comparisons. In figures different letters could be added to describe means differences.

Response: Thanks. We have added the exact p values and different letters to describe means differences in figures.

Line 191-194: Please add the exact p values of the main effect (diet) and of the specific comparisons. In figures different letters could be added to describe means differences.

Response: Thanks. We have added the exact p values and different letters to describe means differences in figures.

Line 201-204: Please add the exact p values of the main effect (diet) and of the specific comparisons. In figures different letters could be added to describe means differences.

Response: Thanks. We have added the exact p values and different letters to describe means differences in figures.